# *Sargassum miyabei Yendo* Brown Algae Exert Anti-Oxidative and Anti-AdipogenicEffects on 3T3-L1 Adipocytes by Downregulating PPARγ

**DOI:** 10.3390/medicina56120634

**Published:** 2020-11-24

**Authors:** Dong Se Kim, Seul Gi Lee, Minyoul Kim, Dongyup Hahn, Sung Keun Jung, Tae Oh Cho, Ju-Ock Nam

**Affiliations:** 1School of Food Science and Biotechnology, Kyungpook National University, Daegu 41566, Korea; aodydirk@naver.com (D.S.K.); lsg100479@knu.ac.kr (S.G.L.); dohahn@knu.ac.kr (D.H.); skjung04@knu.ac.kr (S.K.J.); 2Department of Integrative Biology, Kyungpook National University, Daegu 41566, Korea; rlaalsduf2@naver.com; 3Department of Life Science, Chosun University, Gwangju 501-759, Korea; tocho@chosun.ac.kr; 4Institute of Agricultural Science and Technology, College of Agriculture and Life Sciences, Kyungpook National University, Daegu 41566, Korea

**Keywords:** *Sargassum miyabei Yendo*, brown algae, 3T3-L1, obesity, lipid accumulation

## Abstract

*Background and objectives: Sargassum miyabei Yendo*, belonging to the family Sargassaceae, has been reported to have various biological effects such as anti-tyrosinase activity and anti-inflammation. However, the anti-obesity effect of *Sargassum miyabei Yendo* has not yet been reported. *Materials and Methods:* The effects of *Sargassum miyabei Yendo* extract (SME) on 3T3-L1 adipocytes were screened by3-(4,5)-dimethylthiazo-2-yl-2,5-diphenyltetrazolium bromide (MTT), Oil red O staining, western blot, and Real-time reverse transcription polymerase chain reaction analyses. *Results*: Here, we show that SME had potent 2,2’-azinobis-3-ehtlbezothiazoline-6-sulfonic acid radical decolorization (ABTS) and 2,2-diphenyl-1-picrylhydrazyl (DPPH) antioxidant activity with half maximal inhibitory concentration (IC50) value of 0.2868 ± 0.011 mg/mL and 0.2941 ± 0.014 mg/mL, respectively. In addition, SME significantly suppressed lipid accumulation and differentiation of 3T3-L1 preadipocytes, as shown by Oil Red O staining results. SME attenuated the expression of adipogenic- and lipogenic-related genes such as peroxisome proliferator-activated receptor gamma (PPARγ), CCAAT-enhancer-binding protein alpha (C/EBPα), CCAAT-enhancer-binding protein delta (C/EBPδ), adiponectin, adipose triglyceride lipase (ATGL), fatty acid synthase (FAS), hormone-sensitive lipase (HSL), and lipoprotein lipase (LPL). *Conclusions:* These findings suggest that SME may have therapeutic implications for developing a new anti-obesity agent.

## 1. Introduction

Obesity is defined as excessive weight gain characterized by inordinate fat accumulation in the body [1]. It is associated with energy imbalance, which increases adipocyte hyperplasia, causing hypertrophy of white adipose tissue and increased risk of adverse health conditions, including diabetes, hyperlipidemia, hypertension, and cardiovascular disease [2]. Several studies have reported a relationship between obesity and lipid accumulation by evaluating adipocyte differentiation [3]. The preadipocyte 3T3-L1 cell line is a well-established model for the study of adipogenesis inhibition and various stages of obesity [4]. Many studies have investigated the activity of therapeutic drugs such as synthetic orlistat (Xenical) and sibutramine (Reductil) using the 3T3-L1 cell line, which hasbeen extensively prescribed to treat obesity. However, these drugs are reported to possess adverse effects, including insomnia, steatorrhea, tension headaches, and excessive thirst [5]. Natural sources are deemed safer and less toxic than synthetic agents in therapeutic approaches [6]. Many investigations are thereby focusing on the development of anti-obesity agents from natural sources.

Recent research has shown that some herbs and their metabolites could exert antioxidant activity to mitigate obesity and obesity-related chronic diseases [7]. For instance, green tea, a popular nutraceutical used as an antioxidant, has been reported to increase metabolic rate and body fat metabolism [8].Therefore, plants with antioxidant properties, including green tea, have the potential to exert therapeutic properties against obesity.

Sargassummiyabei is a species of brown algae widespread across the coasts of Japan [9] that is reported to inhibit Reactive oxygen species (ROS) generation and tyrosinase activity [10,11]. Previous studies have found that Sargassum miyabei Yendo extract (SME) could induce anti-inflammatory activity through the inhibition of the nuclear factor kappa-light-chain-enhancer of activated B cells (NF-kB) and mitogen-activated kinase (MAPK) activation [12]. However, specific anti-obesity effects of SME have not been reported. The present study investigated the antioxidant activity and effects of Sargassum miyabei Yendo extract (SME) on lipid accumulation during the differentiation of murine preadipocyte 3T3-L1 cells by measuring levels of adipogenic- and lipogenic-related genes.

## 2. Materials and Methods

### 2.1. Sample Preparation

Sargassummiyabei extract powder (voucher number; MBRB0038TC9306E1) was obtained from the National Marine Biodiversity Institute of Korea (MABIK). We dissolved 20 mg of SargassumMiyabei extract powder in 80% ethanol to a final concentration of 20 mg/mL and then filtered the solution through a 0.2 µM pore size syringe filter. The resulting SME was stored at −20 °C before experimentation.

### 2.2. Chemicals

Dulbecco’s modified Eagle’s medium (DMEM), fetal bovine serum (FBS), and newborn calf serum (BCS) were purchased from Gibco Life Technologies (Grand Island, NY, USA), Insulin, indomethacin, dexamethasone, 3-isobutyl-1methylxanthine, and Oil Red O solution were purchased from Sigma-Aldrich (St Louis, MO, USA). 3-(4,5)-dimethylthiazo-2-yl-2,5-diphenyltetrazolium bromide (MTT) was purchased from Amresco (Solon, OH, USA). Primers were purchased from Macrogen (Seoul, Korea). Anti-C/EBPα antibody and anti-β-actin antibody were purchased from Cell Signaling Technology (Beverly, MA, USA) and Santa Cruz Biotechnology (Santa Cruz, CA, USA), respectively. Anti-PPARγ antibody and anti-adiponectin antibody were purchased from Abcam (Cambridge, UK).

### 2.3. Antioxidant Analysis

#### 2.3.1. 2,2-diphenyl-1-picrylhydrazyl (DPPH)

The antioxidant activity of SME was measured by DPPH (2,2-diphenyl-1-picrylhydrazyl) radical scavenging method [13]. A 100 μL aliquot of SME was mixed with 900 μL DPPH (Sigma-Aldrich Co., St. Louis, MO, USA) solution dissolved in ethanol. The sample was kept in the dark for 30 min, and absorbance was measured by spectrophotometry at 517 nm. DPPH radical scavenging activity was calculated using this formula: DPPH (%) = (control-absorbance − sample absorbance/blank absorbance) × 100

#### 2.3.2. 2,2’-azinobis-3-ehtlbezothiazoline-6-sulfonic Acid Radical Decolorization (ABTS)

ABTS radical solution was prepared by reacting 7 mM ABTS and 2.45 mM aqueous potassium persulfate in 10 mL distilled water for 16 h in the dark at room temperature (RT). A 50 μL aliquot of the sample at different concentrations (0.125, 0.25, 0.5, and 1 mg/mL) was mixed with 950 μL ABTS solution and kept in the dark for 30 min to react. The absorbance was measured by spectrophotometry at 734 nm and calculated as follows:ABTS scavenging activity (%) = (control absorbance − sample absorbance/control absorbance) × 100

### 2.4. Cell Culture

Murine 3T3-L1 preadipocytes cell were purchased from the Korean Cell Line Bank (KCLB, seoul, korea), and 3T3-L1 preadipocytes were maintained in Dulbecco’s modified Eagle’s medium (DMEM) supplemented with 10% BCS and 1% penicillin-streptomycin at 37 °C in a humidified 5% CO_2_ incubator.

### 2.5. Adipocyte Differentiation and Drug Treatment

The culture of 3T3-L1 preadipocytes initiated differentiation into mature adipocytes for 2 days after reaching 100% confluence. It was induced by changing the culture medium to DMEM supplemented with 0.5 mM 3-isobutyl-1-methylxanthine (IBMX), 0.25 µM dexamethasone (DEX), 167 nM insulin, 100 µM indomethacin and 10% FBS for 2 days. After 2 days, the medium was changed to DMEM containing 10% FBS and 10 µg/mL insulin for 8 days, with changes every 2 days. The SME was then filtered through 0.2 µM-pore size syringe filters and dissolved in differentiation media. 3T3-L1 cells were treated every 2 days with SME at concentrations of 100 or 200 µg/mL during the differentiation process.

### 2.6. Cell Viability Assay

3T3-L1 preadipocytes were seeded into 96-well plates at a density of 5 × 10^4^ cells per well. After 24 h of incubation, the cells were treated with SME at concentrations of 100 or 200 µg/mL for 48 h. MTT (3-(4,5)-dimethylthiazo-2-yl-2,5-diphenyltetrazolium bromide) solution was added to each well and incubated for 3 h. The reaction products were dissolved in isopropyl alcohol (Duksan Pure Chemicals, Ansan, Korea) for 1 h. The absorbance of the reaction was measured at 595 nm.

### 2.7. Oil Red O Staining

Oil Red O staining (ORO) was performed as previously described [14,15]. 3T3-L1 preadipocytes were induced to differentiate in SME’s presence or absence (100 or 200 µg/mL). 3T3-L1 preadipocytes and differentiated adipocytes were washed with phosphate-buffered saline (PBS), fixed with 4% formaldehyde for 1 h, and stained with filtered 0.3% Oil Red O solution (Sigma-Aldrich, St. Louis, MO, USA) at RT for 15 min. Then, stained cells were washed three times with distilled water and photographed with a microscope at 400× magnification. The amount of lipid accumulation was quantified by staining the cells with Oil Red O dissolved in isopropyl alcohol and measuring the absorbance at 450 nm.

### 2.8. Triglyceride Assay

The triglyceride (TG) assay was performed as previously described [16]. Briefly, 3T3-L1 preadipocytes were seeded into 6-well plates with differentiation media and SME (100 and 200 µg/mL) for up to 8 days. After differentiation, cells were lysed in 5% NP-40 lysis buffer, and triglycerides were converted to fatty acids and glycerol with a lipase enzyme. The absorbance of the released glycerol was measured at 570 nm.

### 2.9. Real-Time Reverse Transcription Polymerase Chain Reaction (RT-PCR)

After differentiation of 3T3-L1 cells with SME (100 or 200 µg/mL), total RNA was extracted from adipocytes using Trizol reagent. A cDNA library was synthesized with the PrimeScriptTM RT Reagent Kit (TaKaRa Bio, Kyoto, Japan). mRNA expression levels were quantified by analysis of cDNA implemented by an iCycleriQTM Real-Time PCR Detection System (Bio-Rad Laboratories, Hercules, CA, USA) using SYBR Green (TOYOBO, Japan). The conditions of the PCR reaction were as follows: a denaturation cycle at 95 °C for 10 min, followed by 95° C for 15 sec and 60 °C for 10 min. mRNA expression levels were normalized to β-actin, and relative gene expression was expressed as fold change in mRNA expression level. The sequences of PCR primers used in the experiment are shown in Table 1.

### 2.10. Western Blot Assay

After 3T3-L1 preadipocytes were differentiated in the absence or presence of SME (100 and 200 µg/mL), cells were lysed with radioimmunoprecipitation assay (RIPA) lysis buffer containing phosphatase inhibitor and protease inhibitor. Total protein (30 µg) content was separated on a 10% sodium dodecyl sulfate-polyacrylamide gel electrophoresis (SDS)-polyacrylamide gel via electrophoresis and transferred to a nitrocellulose membrane. The membranes were blocked with 5% skim milk in PBS for 1 h and incubated with the relevant primary antibody overnight at 4 °C, after which the membranes were washed five times with 1× Tris buffered saline-tween (TBS-T) buffer. The membranes were then incubated with horseradish peroxidase (HRP)-conjugated secondary antibody for 1 h and washed five times with TBS-T buffer. Protein bands were visualized using electrochemiluminescence (ECL) and detected on a Fusion Solo Detector (VilberLourmat, Marne La Vallee, France). The total amount of protein was assessed by evaluating protein bands using Image-J software and normalizing values to β-actin.

### 2.11. Statistical Analysis

Data are represented as means ± standard error of the mean (SEM) and standard deviation (SD) and were analyzed using one-way analysis of variance (ANOVA). A *p*-value of less than 0.01 was considered statistically significant. 

## 3. Results

This section may be divided by subheadings. It should provide a concise and precise description of the experimental results, their interpretation, and the experimental conclusions.

### 3.1. Antioxidant Activity

We evaluated SME’s antioxidant properties by assessing ABTS radical cation and DPPH radical scavenging activity (Table 2). The half maximal inhibitory concentration (IC50) values of SME for scavenging ABTS and DPPH were 0.2868 mg/mL and 0.2941 mg/mL, respectively. These results suggested that SME showed potent antioxidant activity in ABTS and DPPH scavenging activity, although the effects were lower than IC50 values of ascorbic acid.

### 3.2. Effects of SME on 3T3-L1 Preadipocyte Cell Viability

Before evaluating SME’s effects on 3T3-L1 cells, we first examined whether SME was cytotoxic to 3T3-L1 preadipocytes to determine a suitable experimental concentration. Results showed that SME was not cytotoxic to preadipocytes at a concentration of 200 µg/mL (Figure 1). We thereby used SME at 100 and 200 µg/mL for subsequent experiments.

### 3.3. SME Effects on Lipid Accumulation and Triglyceride Composition in 3T3-L1 Adipocytes

We investigated the effects of SME on adipogenesis by measuring intracellular lipid accumulation and triglyceride contents; mature adipocytes are characterized by the presence of lipid droplets, which are not found in preadipocytes. SME significantly inhibited adipocyte differentiation and intracellular lipid accumulation in a dose-dependent manner (Figure 2A,B). Furthermore, SME significantly decreased the intracellular triglyceride contents compared with control cells (Figure 2C). These results indicate that SME could exert potent anti-adipogenic effects as evidenced by decreased intracellular lipid accumulation.

### 3.4. Effects of SME on Adipogenesis-Related Gene Expressions during 3T3-L1 Cell Differentiation

We investigated the molecular mechanisms underlying SME’s anti-adipogenic properties on adipocyte differentiation by examining the expression levels of three adipogenesis-related genes, PPARγ, C/EBPα, and adiponectin. SME treatment was associated with a significant decrease in mRNA expression of PPARγ, C/EBPα, and adiponectin (Figure 3A). Adipocyte differentiation is a complex process accompanied by coordinated changes in the expression of genes [17]. C/EBPδ is a critical determinant for the early stage of adipocyte differentiation and later activates the expression of key adipogenic regulators such as PPARγ and C/EBPα [18,19]. Therefore, weinvestigated SME’s effect on the mRNA expression level of the early adipogenic transcription factor C/EBPδat an early time point of differentiation. We observed that SME treatment significantly decreased the mRNA expression of C/EBPδin a concentration-dependent manner (Figure 3B).Protein expression levels of PPARγ, C/EBPα, and adiponectin were also decreased in a dose-dependent manner, concomitant with mRNA expression patterns (Figure 3C,D). These results suggest that SME inhibited adipogenesis by downregulating the PPARγ signaling pathway, and the inhibitory effect may also begin in the early stage of adipocyte differentiation.

### 3.5. Effects of SME on Lipogenic-Related Gene Expressions on 3T3-L1 Adipocytes

To elucidate the molecular mechanisms involved in suppressing intracellular lipid accumulation in SME-treated 3T3-L1 adipocytes, we explored the effect of SME on the mRNA expression of lipogenic-related genes, including adipose triglyceride lipase (ATGL), fatty acid synthase (FAS), hormone-sensitive lipase (HSL), and lipoprotein lipase (LPL). We found that SME treatment significantly decreased the mRNA expression levels of ATGL, fatty acid synthase (FAS), hormone-sensitive lipase (HSL), and lipoprotein lipase (LPL) in a dose-dependent manner (Figure 4). These results suggest that SME inhibited lipid biosynthesis induction in 3T3-L1 by regulating the expression of the primary lipogenic-related genes.

## 4. Discussion

SME is known to contain various carotenoids, important dietary nutrients with potential antioxidant activity, such as neoxanthin, fucoxanthin, and fucoxanthinol [20]. Studies have recently demonstrated that dietary supplements with radical scavenging activity could reduce body weight and amelioration of obesity-related disorders [21,22]. Many studies have also suggested that oxidative stress may be an important link between obesity and obesity-related diseases [13]. In this study, SME presented significant antioxidant activity in ABTS and DPPH assays, implying that SME’s anti-obesity properties could be associated with carotenoid components with antioxidant function present in the extract [23]. However, it should be noted that direct evidence concerning whether the effects of SME were associated with major carotenoids was not assessed in this study.

In the present study, we found that SME inhibited intracellular lipid accumulation and adipocyte differentiation in 3T3-L1 cells without cytotoxicity. Multiple studies have reported that adipogenesis-related genes’ expression was significantly elevated in murine models of obesity [24]. In 3T3-L1 adipocytes, the PPARγ signaling pathway has been identified to facilitate cell enlargement and intracellular triglyceride accumulation [25]. We demonstrated that SME treatment significantly attenuated the expression of PPARγ, C/EBPα, C/EBPδ, and adiponectin compared to that in control cells. In addition, we showed that SME treatment regulated the mRNA expression of lipogenic-related genes. These results imply that SME suppressed adipogenesis and lipid accumulation by regulating the complex transcriptional cascade, which is related to several stages of differentiation on 3T3-L1 adipocytes.Our findings suggest SME as a promising natural source for treating obesity. To the best of our knowledge, these results represent the first evidence of SME’s anti-adipogenic effects on 3T3-L1 adipocytes. However, further research to elucidate this capacity and its underlying therapeutic properties in an obese animal model is warranted.

## 5. Conclusions

We found that SME exhibited measurable antioxidant activity in ABTS and DPPH assays. We also demonstrated that SME treatment was associated with significant inhibition of 3T3-L1 cell differentiation, as evidenced by a visible reduction in lipid accumulation. Our finding revealed that SME inhibited adipogenesis by mediating the downregulation of the expression of adipogenic- and lipogenic-related genes suggesting that SME may be a promising natural source for the treatment of obesity.

## Figures and Tables

**Figure 1 medicina-56-00634-f001:**
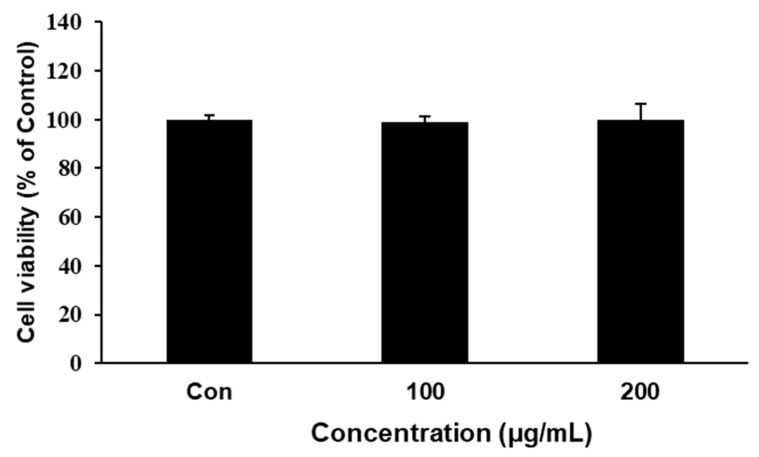
Effect of *Sargassum miyabei Yendo* extract (SME) on the cell viability of 3T3-L1 preadipocytes. Cell viabilities were determined by 3-(4,5)-dimethylthiazo-2-yl-2,5-diphenyltetrazolium bromide (MTT) assay. 3T3-L1 preadipocytes were treated with SME at 100 or 200 µg/mL for 48 h. Formed formazan was quantified by measuring the absorbance at 595 nm. Each experiment was repeated in triplicate. Bars represent mean ± SD from three independent experiments. Con: positive controls.

**Figure 2 medicina-56-00634-f002:**
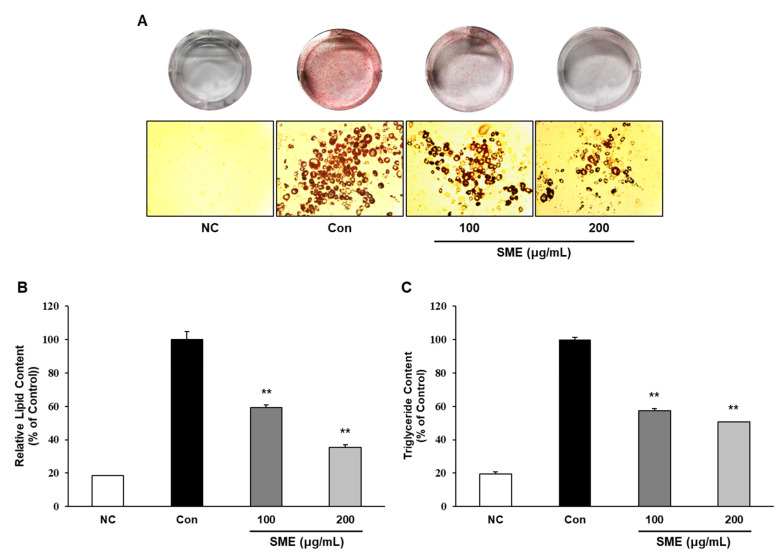
Effects of SME on lipid accumulation and triglyceride content during 3T3-L1 cell differentiation. 3T3-L1 preadipocytes were treated with differentiation medium in the presence or absence of SME at 100 or 200 μg/mL for 8 days. Lipid accumulation was quantified by staining with Oil Red O solution, and absorbance was measured at 495 nm. (**A**) Cells photographed at 400× magnification after differentiation. (**B**) Quantification of stained intracellular lipid content (**C**) Intracellular triglyceride contents, quantified by triglyceride assay kit and assessed at OD570. Preadipocytes were used as negative controls (NC), and fully differentiated adipocytes were used as positive controls (Con). Each experiment was repeated in triplicate. Bars represent mean ± SD. ** *p*< 0.01 compared to control.

**Figure 3 medicina-56-00634-f003:**
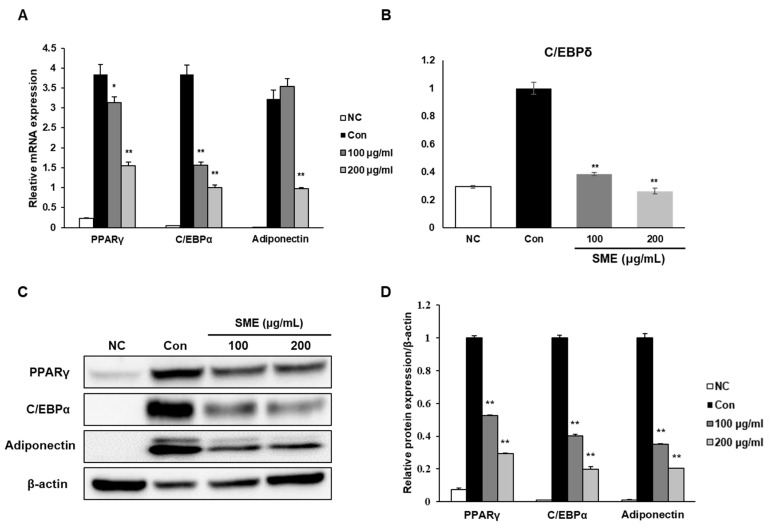
Effect of SME on adipogenesis-related genes in 3T3-L1 adipocytes.3T3-L1 preadipocytes were cultured in differentiation medium in the presence or absence of SME at 100 or 200 μg/mL during differentiation for 8 days. RNA and proteins were isolated from the adipocytes following differentiation. (**A**) The relative mRNA expression levels of proliferator-activated receptor-γ (PPARγ), CCAAT/enhancer-binding protein α (C/EBPα), and adiponectin were assessed by real-time PCR. (**B**) 3T3-L1 adipocytecells were cultured in a differentiation medium with SME at the indicated concentrations for a day. After treatment, the related mRNA expression levels ofCCAAT-enhancer-binding protein delta (C/EBPδ) were determined by real-time PCR. (**C**) Representative images of Western blot results.(**D**) Quantification of protein expression levels. Each experiment was repeated in triplicate. Bars represent mean ± SD.* *p* < 0.05 and ** *p* < 0.01 compared to control group.

**Figure 4 medicina-56-00634-f004:**
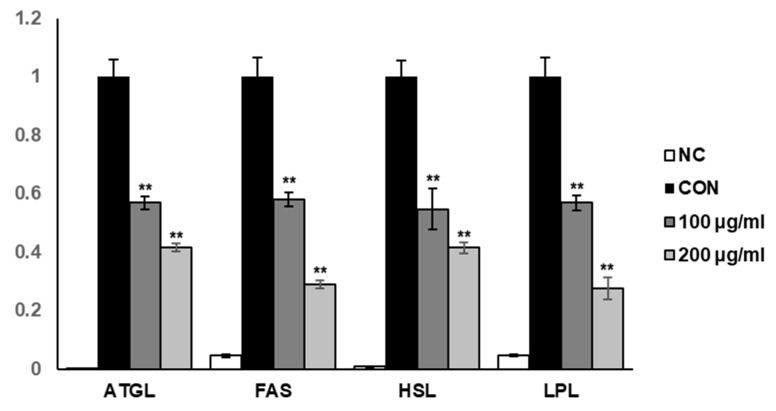
Effects of SME on the expression of lipogenic-related genes in 3T3-L1 adipocytes.3T3-L1 preadipocytes were cultured in differentiation medium with SME at the indicated concentrations for 8 days. The mRNA expressions of adipose triglyceride lipase (ATGL), fatty acid synthase (FAS), hormone-sensitive lipase (HSL), and lipoprotein lipase (LPL) were determined by real-time PCR. Each experiment was repeated in triplicate. Bars represent mean ± SD. ** *p* < 0.01 compared with the control group.

**Table 1 medicina-56-00634-t001:** sequences used for quantitative RT-PCR.

Gene Name	Accession No.	Forward Primer	Reverse Primer
Adiponectin	NM_009605.4	GATGGCACTCCTGGAGAGAA	CGTGCGTGACATCAAAGAGAA
ATGL	NM_001163689.1	CAACGCCACTCACATCTACGG	GGACACCTCAATAATGTTGGCAC
C/EBPα	NM_001287523.1	CAAGAACAGCAACGAGTACCG	GTCACTGGTCAACTCCAGCAC
C/EBPδ	X62600.1	TCCACGACTCCTGCCATGTAC	AAGAGTTCGTCGTGGCACAG
FAS	AF127033.1	GGTCGTTTCTCCATTAAATTCTCAT	CTAGAAACTTTCCCAGAAATCTTCC
HSL	NM_001039507.2	CAGAAGGCACTAGGCGTGATG	GGGCTTGCGTCCACTTAGTTC
LPL	NM_008509.2	CTGGTGGGAAATGATGTGG	TGGACGTTGTCTAGGGGGTA
PPARγ	AB644275.1	GGAAGACCACTCGCATTCCTT	GTAATCAGCAACCATTGGGTCA
β-actin	NM_007393.4	CGTGCGTGACATCAAAGAGAA	GCTCGTTGCCAATAGTGATGA

RT-PCR: Reverse Transcription PCR; ATGL: Adipose triglyceride lipase; C/EBPα: CCAAT/enhancer-binding proteinα; C/EBPδ: CCAAT/enhancer-binding protein δ; FAS: Fatty acid synthase; HSL: Hormone-sensitive lipase; LPL: Lipoprotein lipase; PPARγ: Peroxisome proliferator-activated receptorγ.

**Table 2 medicina-56-00634-t002:** Antioxidant activity of SME in ABTS and DPPH assays.

		IC50 (mg/mL)
ABTS assay	Ascorbic acid (standard)	0.0622 ± 0.002
	SME	0.2868 ± 0.011
DPPH assay	Ascorbic acid (standard)	0.0678 ± 0.003
	SME	0.2941 ± 0.014

SME: Sargassum miyabei Yendo extract; ABTS: 2,2’-azino-bis(3-ethylbenzothiazoline-6-sulfonic acid; DPPH: 2,2-diphenyl-1-picryl-hydrazyl-hydrate; IC50: half maximal inhibitory concentration.

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
