# Peer review of "Sargassum miyabei Yendo Brown Algae Exert Anti-Oxidative and Anti-AdipogenicEffects on 3T3-L1 Adipocytes by Downregulating PPARγ"

_medicina, 2020, doi:10.3390/medicina56120634_

Round 1
Reviewer 1 Report
The manuscript "Sargassum miyabei Yendo brown algae exert anti-oxidative and anti-adipogenic effects on 3T3-L1 adipocytes by downregulating PPARy" submitted by J.-O. Nam and coworkers investigates the action of this natural product on the differentiation and lipid synthesis in cultured adipocytes.
The study may be of interest for researchers engaged in the areas of metabolic diseases including obesity and their therapy provided the following points have been adequately addressed by the authors.
Major points
- According to the demonstrated IC50 for the anti-oxidative action of the extract (about 0.3 mg/ml) it seems unlikely that this activity is responsible for the effects on differentiation and gene expression considerable since the concentrations used in thses assays are considerably below the IC50 values. In consequence, the causal relationship between these activities remains to be established.
- It is unclear from the experimental design (assaying lipid accumulation by Oil Red staining or determination of lipid content) whether the extract actually interferes with the differentiation process or with the lipid biosynthetic pathways (e.g. by inhibition of an enzyme). To clarify this important point, the de novo lipid biosynthesis (lipogenesis) has to be determined with differentiated (primary or cultured) adipocytes.
- The effect of the extract on cell viability was studied after incubation for 2 days (outcome no effect up to 0.3 mg/ml). However, during differentiation the cells were incubated with the extract for 8 days (with changes every 2 days). Thus according to this protocol, it cannot be excluded that (part of) the downregulation of differentiation is actually due to impairment of the viabiliy of the differentiating adipocytes during day 6 to 8.
- Along this argumentation it would be interesting to study the period/time point required/critical for the initiation of the differentiation blockade.
Minor point
Some spelling errors, e.g.
Line 30: "White" for "shite"
Line 224: "To the best of" instead of "to be best"
Author Response
Dear Reviewer,
The authors of this paper appreciate your intensive review of our paper. Your comments have been very valuable and helpful for revising and improving our paper. Based on your guidance, our revision includes a number of positive changes as follows:
COMMENT 1: The manuscript “Sargassum miyabei Yendo brown algae exert anti-oxidative and anti-adipogenic effects on 3T3-L1 adipocytes by downregulating PPARy” submitted by J.-O. Nam and coworkers investigates the action of this natural product on the differentiation and lipid synthesis in cultured adipocytes.
The study may be of interest for researchers engaged in the areas of metabolic diseases including obesity and their therapy provided the following points have been adequately addressed by the authors.
According to the demonstrated IC50 for the anti-oxidative action of the extract (about 0.3 mg/ml) it seems unlikely that this activity is responsible for the effects on differentiation and gene expression considerable since the concentrations used in thses assays are considerably below the IC50 values. In consequence, the causal relationship between these activities remains to be established.
RESPONSE: Thank you very much for your comment. In this study, we revealed that SME exhibited both anti-oxidant and anti-differentiation effects and that the effective concentration was significantly different between the two effects. We could hypothesize that the calculated effective concentrations would depend on the experimental system. There would be a difference because the anti-oxidant effect of SME was assessed by enzyme assay, which is a non-cell based experiment, while the anti-differentiation effect was evaluated using 3T3-L1 cell-based experiments. Nevertheless, many antioxidant plants (e.g., green tea) have the potential to exert the therapeutic effects against obesity. In other words, there is a strong relationship between anti-oxidant and anti-obesity effects. We have addressed the relationship between two effects in the Introduction section.
COMMENT 2: It is unclear from the experimental design (assaying lipid accumulation by Oil Red staining or determination of lipid content) whether the extract actually interferes with the differentiation process or with the lipid biosynthetic pathways (e.g. by inhibition of an enzyme). To clarify this important point, the de novo lipid biosynthesis (lipogenesis) has to be determined with differentiated (primary or cultured) adipocytes.
RESPONSE: Thank you very much for your comment. In addition to the adipogenic-related genes (PPARγ, C/EBPα, and adiponectin), we examined the effect of SME on the mRNA expression of lipogenic-related genes, including adipose triglyceride lipase (ATGL), fatty acid synthase (FAS), hormone-sensitive lipase (HSL), and lipoprotein lipase (LPL). As expected, we found that SME treatment significantly suppressed the mRNA expression of ATGL, FAS, HSL, and LPL in a dose-dependent manner. We have added these data to the Results section (3.5) and Figure 4.
COMMENT 3: The effect of the extract on cell viability was studied after incubation for 2 days (outcome no effect up to 0.3 mg/ml). However, during differentiation the cells were incubated with the extract for 8 days (with changes every 2 days). Thus according to this protocol, it cannot be excluded that (part of) the downregulation of differentiation is actually due to impairment of the viabiliy of the differentiating adipocytes during day 6 to 8.
RESPONSE: Thank you very much for your comment. Although we cannot prove whether long-term treatment with SME would affect the viability of 3T3-L1, we had visually confirmed no SME cytotoxicity at day 8 of treatment of the 3T3-L1 adipocytes with SME. We would be appreciative if the reviewer understands that it is a too short a time to demonstrate no cytotoxicity of SME with long-term treatment.
COMMENT 4: Along this argumentation it would be interesting to study the period/time point required/critical for the initiation of the differentiation blockade.
RESPONSE: Thank you very much for your suggestion. We elucidated the effect of SME on the mRNA expression level of the early adipogenic transcription factor C/EBPδ at an early time in differentiation. The SME treatment significantly decreased the mRNA expression of C/EBPδ in a dose-dependent manner. This result implies that the inhibitory effects of SME may begin in an early stage of adipocyte differentiation. We have added these data to the Results section (3.4) and Figure 3B.
Minor point
COMMENT 5: Some spelling errors, e.g.
Line 30: “White” for “shite”
Line 224: “To the best of” instead of “to be best”
RESPONSE: We apologize for the mistakes and have corrected the misspelled words.

Reviewer 2 Report
The manuscript by Dong Se Kim and colleagues describes the potential of Sargassum miyabei Yendo extract (SME) as a potential antioxidant and anti-adipogenesis regulator. The entire study is carried out using 3T3-L1 murine pre-adipocytes cells as model of adypocite differentiation. Cells were treated with different doses of SME and differentiated to adypocites. SMEresulted to be not toxic and able to limit triglyceride accumulation, inhibiting important adipogenesis-related players of PPARy, C/EBPα, and adiponectin (both mRNA and protein levels).
The work is linear, simple and properly describes what the authors made. There is no biological mechanism explaining the presented data, however I think this would be beyond this study.
I appreciate the authors suggesting further investigations about the role of SME in adypocite differentiation.
Author Response
Dear Reviewer,
The authors appreciate your intensive review of our paper. Your comment is valuable and very helpful for the revision and improvement of our paper. Based on your guidance, we have added data as follows:
Reviewer 2
The manuscript by Dong Se Kim and colleagues describes the potential of Sargassum miyabei Yendo extract (SME) as a potential antioxidant and anti-adipogenesis regulator. The entire study is carried out using 3T3-L1 murine pre-adipocytes cells as model of adypocite differentiation. Cells were treated with different doses of SME and differentiated to adypocites. SME resulted to be not toxic and able to limit triglyceride accumulation, inhibiting important adipogenesis-related players of PPARy, C/EBPα, and adiponectin (both mRNA and protein levels).
The work is linear, simple and properly describes what the authors made. There is no biological mechanism explaining the presented data, however I think this would be beyond this study.
I appreciate the authors suggesting further investigations about the role of SME in adypocite differentiation.
RESPONSE: Thank you very much for your suggestion. Adipocyte differentiation is a complex process accompanied by coordinated changes in gene expression (Cho et al., 2004). C/EBPδ is a critical determinant for the early stage of adipocyte differentiation and later activates the expression of key adipogenic regulators such as PPARγ and C/EBPα (Lee et al., 2017 and Hishida et al., 2009). Given the contribution of C/EBPδ to the differentiation of adipocytes, we elucidated the effect of SME on the mRNA expression level of the early adipogenic transcription factor C/EBPδ at an early time in differentiation (day 1). SEM treatment significantly decreased the mRNA expression of C/EBPδ genes in a dose-dependent manner. This result implies that the inhibitory effect of SME may begin at an early stage of adipocyte differentiation. We have added these data to the Results section (3.4) and Figure. 3B.
- Cho, H.J.; Park, J.; Lee, H.W.; Lee, Y.S.; Kim, J.B. Regulation of adipocyte differentiation and insulin action with rapamycin. Biochemical and biophysical research communications 2004, 321, 942-948.
- Lee, S.G.; Taeg, K.K.; Nam, J.O. Silibinin Inhibits Adipogenesis and Induces Apoptosis in 3T3-L1 Adipocytes. Microbiology and Biotechnology letters 2017, 45, 27~34 doi:https://doi.org/10.4014/mbl.1610.10005.
- Hishida, T.; Nishizuka, M.; Osada, S.; Imagawa, M. The role of C/EBPδ in the early stages of adipogenesis. Biochimie 2009, 91, 654-657.

Round 2
Reviewer 1 Report
I appreciate the inclusion of additional experimental data in the revised version.
Thereby the authors have met the majority of my criticism (albeit not all, see comment 3) and improved the manuscript considerably.